# Roles of miR-640 and Zinc Finger Protein 91 (ZFP91) in Angiopoietin-1-Induced In Vitro Angiogenesis

**DOI:** 10.3390/cells9071602

**Published:** 2020-07-02

**Authors:** Sharon Harel, Veronica Sanchez-Gonzalez, Raquel Echavarria, Dominique Mayaki, Sabah NA Hussain

**Affiliations:** 1Department of Critical Care, McGill University Health Centre, Montreal, QC H4A 3J1, Canada; sharon.harel@mail.mcgill.ca (S.H.); veronica.sanchez@mail.mcgill.ca (V.S.-G.); raquel.echavarria@mail.mcgill.ca (R.E.); dominique.mayaki@muhc.mcgill.ca (D.M.); 2Meakins-Christie Laboratories, Department of Medicine, McGill University, Montreal, QC H4A 3J1, Canada

**Keywords:** angiogenesis, miRNAs, angiopoietins, endothelial cells

## Abstract

Angiopoietin-1 (Ang-1) is a ligand of Tie-2 receptors that promotes angiogenesis. It has been established that regulatory loops exist between angiogenic growth factors and distinct pro or anti-angiogenic miRNAs, but the nature and the roles of Ang-1-regulated miRNAs remain unclear. In this study, we assessed the role of miR-640 in Ang-1-induced angiogenesis in human umbilical vein endothelial cells (HUVECs). Exposure to Ang-1 (300 ng/mL) from 6 to 72 h significantly decreased expression of mature miR-640, a response that was mediated by Tie-2 receptors and was also observed in response to Ang-2, the vascular endothelial growth factor, and transforming growth factor β. Increasing miR-640 levels using a mimic inhibited Ang-1-induced cell migration and capillary-like tube formation whereas inhibition of miR-640 enhanced these responses. Pull down assays of biotinylated miR-640 revealed that miR-640 directly targets Zinc Finger Protein 91 (ZFP91), an atypical E3-ubiquitin ligase. Ang-1 exposure induced ZFP91 expression through down-regulation of miR-640. Silencing of ZFP91 significantly inhibited Ang-1-induced cell migration and tube formation. We conclude that Ang-1 upregulates ZFP91 expression through transcriptional down-regulation of miR-640 and that ZFP91 plays important roles in the promotion of Ang-1-induced endothelial cell migration and differentiation.

## 1. Introduction

Angiopoietin-1 (Ang-1) and its receptor, Tie-2, are major regulators of physiological and pathological angiogenesis. In mouse embryos, knocking down Ang-1 gene is associated with major vascular abnormalities [1]. In cultured endothelial cells (ECs), Ang-1 inhibits inflammation and apoptosis and stimulates differentiation, sprouting, and migration [2]. In adult animals, Ang-1 stimulates vascular remodeling and wound healing, increases lymphangiogenesis [3,4,5], and inhibits apoptosis [6]. It also promotes EC migration and differentiation, and attenuates permeability [7,8,9]. The Ang-1/Tie-2 axis stimulates the PI-3 kinase and three members of the mitogen activated protein kinases (MAPKs; Erk1/2, p38, and SAPK/JNK) pathways [2,10] and these pathways mediate the majority of angiogenic properties of this axis. Despite a major progress in our understanding of Ang-1/Tie-2 signaling, little information is published regarding the importance of non-coding RNAs in general and micro RNAs (miRNAs) in particular in the signaling and angiogenic properties of the Ang-1/Tie-2 axis.

It is increasingly evident that miRNAs play important roles in angiogenesis. Angiogenesis is strongly inhibited when Dicer and Drosha, enzymes that are essential to miRNA biogenesis, are deleted in mice [11]. Angiogenesis-related miRNAs are either pro-angiogenic or anti-angiogenic. For example, miR-126 is an abundantly expressed pro-angiogenic miRNA that promotes vascular endothelial cell growth factor (VEGF) signaling, maintains vascular integrity, and enhances EC migration and proliferation [12,13]. By comparison, the miR-17-92 cluster is an example of anti-angiogenic miRNAs and its members attenuate EC sprouting and suppress angiogenesis [14]. To date, little information is available about the roles of miRNAs on the biological effects of Ang-1. Recently, we reported that miR-146b-5p promotes the inhibitory effects of the Ang-1/Tie-2 axis on inflammation in ECs by selectively targeting key elements of the toll-like receptor 4 signaling pathway [15]. More recently, we identified six novel miRNAs (miR-575, miR-103b, miR-330-5p, miR-1287-5p, miR-557, and miR-1468-5p) as negative regulators of EC survival, proliferation, migration, and differentiation and that the expression of these miRNAs are decreased by the Ang-1/Tie-2 axis in ECs [16].

We recently conducted pilot experiments to identify miRNAs whose expression in human umbilical vein endothelial cells (HUVECs) are downregulated in response to 24 h exposure to Ang-1. These experiments indicated that Ang-1 exposure elicits a significant decrease in the expression of miR-640. Little information is available regarding the expression and functional roles of miR-640 in various cells. To our knowledge, Zhou et al. [17] is the only group to document the existence of miR-640 in ECs. They reported that exposure of ECs to hydrogen sulfide decreases miR-640 expression and that overexpression of miR-640 inhibits hydrogen sulfide-induced EC migration and differentiation. Whether miR-640 plays a significant role in regulating angiogenic processes elicited by angiogenesis factors in general, and by Ang-1 in particular, remains unclear. The main goal of this study was to assess the regulation of miR-640 expression and its function in regulating Ang-1-induced in-vitro angiogenesis. We also aimed at identifying the targets through which miR-640 regulates angiogenesis. We hypothesized that miR-640 functions as an anti-angiogenic miRNA and that it inhibits angiogenesis by selectively targeting zinc finger protein 91 (ZFP91). We also hypothesized that ZFP91 is required for the pro-angiogenic effects of the Ang-1/Tie-2 axis.

## 2. Material and Methods

### 2.1. Materials

Recombinant human Ang-1 and Ang-2 were purchased from R&D Systems (Minneapolis, MN, USA). Vascular endothelial growth factor (VEGF), fibroblast growth factor 2 (FGF-2), and transforming growth factor beta (TGF-β) were purchased from Bioshop (Burlington, ON, USA). Antibodies for zinc finger protein 91 (ZFP91) and β-ACTIN were purchased from Sigma-Aldrich (Oakville, ON, USA). Cleaved caspase-3 antibody was purchased from Cell Signaling Technology (Danvers, MA, USA). Human umbilical vein endothelial cells (HUVECs) were purchased from Lonza (Basel, Switzerland).

### 2.2. Cell Culture

HUVECs were maintained in complete MCDB 131 medium (Life Technologies, Rockville, MD, USA) supplemented with 20% fetal bovine serum (FBS), EC growth supplement, 2 mM glutamine, heparin, and gentamicin sulfate, and incubated at 37 °C and 5% CO_2_. In experiments where EC migration, survival, and activation of caspase-3 activity were measured, cells were cultured in basal MCDB131 medium containing gentamicin sulfate and 2% FBS.

### 2.3. Regulation of miR-640 Expression

HUVECs were cultured for 6 h in basal medium and were then maintained for 2, 4, 6, 12, 24, 48, and 72 h in basal medium containing aliquots of Ang-1 (300 ng/mL) or phosphate buffered saline PBS (control). To assess whether other angiogenesis factors regulate miRNA expression, we exposed HUVECs to VEGF (40 ng/mL), FGF-2 (10 ng/mL), TGF-β (2 ng/mL), or Ang-2 (300 ng/mL) for 24 h. Total RNA was extracted and miRNA expression was detected as described below.

### 2.4. miRNA Analysis

Total RNA was extracted using Qiazol^®^ (Qiagen, Germantown, MD, USA) and miRNeasy Mini Kit (Qiagen, Hilden, Germany). miRNAs were detected using an NCode™ miRNA qRT-PCR Kit (Invitrogen Inc. Thermo Fisher Scientific, Waltham, MA, USA), and real-time PCR using specific primers, SYBR^®^ green (Life Technologies Corporation, Carlsbad, CA, USA), and a 7500 Real-Time PCR System (Applied Biosystems, Foster City, CA, USA; Appendix A). Pri-miR-640 was measured using specific TaqMan^®^ assays (Applied Biosystems). All experiments were performed in triplicate. Relative miRNA expression was calculated on the basis of the C_T_ method where C_T_ values of individual miRNA data were normalized to C_T_ values of U6 snRNA as previously described [18]. 

### 2.5. Adenoviral Infection

When HUVECS reached 60–70% confluency they were then exposed for 6 h to a serum-free medium containing 100 multiplicity of infection (MOI) virus units of adenoviruses expressing GFP or EX-TEK (Vector Biolabs, Philadelphia, PA, USA) [19]. Cells were then allowed a 48 h recovery period prior to exposure to PBS or Ang-1.

### 2.6. Exosome Isolation

HUVECs were cultured in complete MCDB131 medium until they reached full confluence. The medium was then changed to basal MCDB131 medium for 6 h, cells were then exposed to basal medium containing aliquots of Ang-1 (300 ng/mL) or PBS for 24 h. The Exiqon miRCURY™ Exosome Isolation Kit (Woburn, MA, USA) was used to isolate exosomes secreted from cells into media. Total RNA was extracted from isolated exosomes and miRNA expression was measured as described below. 

### 2.7. Transfection with siRNA, miRNA Mimics, and LNA-Inhibitors

HUVECs were transfected with synthetic siRNA oligos (10 nM of scrambled or specific to ZFP91), synthetic mature miRNA mimic (15 nM, Ambion Inc., Austin, TX, USA), or LNA- miRNA Inhibitor (25 nM, Exiqon Inc., Vedbaek, Denmark) using Lipofectamine™ RNAiMAX (Invitrogen Inc.) according to the manufacturers’ instructions. After 48 h of recovery in complete MCDB131 medium, cells underwent specific experimental procedures. 

### 2.8. Cell Number and Cleaved Caspase-3 Detection

HUVECs transfected with siRNA oligos, miRNA mimics, or miRNA inhibitors were seeded onto 12-well plates (8 × 10^4^/cm^2^). Cells were cultured for 24 h in complete medium or basal medium containing aliquots of PBS or Ang-1 (300 ng/mL). Cells were then counted using a hemocytometer and underwent immunoblotting for cleaved caspase-3 measurements.

### 2.9. Cell Migration

Cell migration was evaluated using a scratch (wound) healing assay as in our previous study [20]. In brief, HUVEC monolayers were wounded with a 200 μL pipette tip. Cells were then maintained for 8 h in basal medium containing PBS or Ang-1 (300 ng/mL). Wounded areas were visualized with an Olympus inverted microscope and quantified using Image-Pro Plus™ 8.0 software (Media Cybernetics, Bethesda, MD, USA). Values are reported as percent wound healing, which were calculated using the following formula:Percent wound healing = [1 − (wound area at t8/wound area at t0)] × 100(1)
where t8 is the time (8 h) over which cells were maintained in the medium and t0 is the time immediately after wounding.

### 2.10. Capillary-Like Tube Formation

Cells transfected with siRNA-, miRNA mimic-, or inhibitor-transfected were seeded onto 96-well plates pre-coated with growth factor-reduced Matrigel^®^ (Corning Life Sciences, Corning, NY, USA) (1 × 10^4^ cells per well). Cells were cultured for 24 h in basal medium containing PBS (control) or Ang-1 (300 ng/mL). Olympus inverted microscope (40×) was used to capture whole-well images at 40× magnification. Images were analyzed using Image-Pro Plus™ software. Angiogenic tube formation was determined by counting total branches in each field, as previously described [21].

### 2.11. mRNA Analysis

Total RNA was extracted using PureLink RNA mini kit (Life Technologies, Inc.) according to the manufacturer’s protocol. mRNA levels were detected with real-time PCR using specific primers, SYBR^®^ green, and a 7500 Real-Time PCR System (Appendix A). GAPDH and β-ACTIN were used as control genes. All experiments were performed in triplicate. Relative mRNA expression was determined using the C_T_ method (2^−ΔΔCT^) as previously described [22].

### 2.12. Immunoblotting

HUVECs were lysed in RIPA buffer (Santa Cruz Biotechnologies) and protein concentration was measured with the Bio-Rad assay using bovine serum albumin (BSA) as a standard. Cell lysates were diluted in Laemmli sample buffer, boiled for 5 min, and loaded onto Tris-glycine SDS-polyacrylamide gels. After SDS-PAGE, the proteins were transferred onto polyvinylidene difluoride (PVDF) membranes, blocked with 5% non-fat dry milk and then incubated with the specific primary antibodies overnight at 4 °C. Proteins were detected using horseradish peroxidase-conjugated secondary antibodies and ECL reagents (Chemicon, Temecula, CA, USA). 

### 2.13. 3′UTR Luciferase Constructs

pEZX-MT06^™^ dual reporter plasmids containing 2000 bp of wild-type (wt) ZFP91 3’UTR cloned downstream of a SV40-driven Firefly Luciferase cassette and a CMV-driven Renilla Luciferase (Genecopeia, Rockville, MD, USA). The 2000 bp region of ZFP91 3’UTR was predicted to have two miR-640 binding regions (Appendix A). The mutated versions of these two regions (region1-mut and region2-mut) carrying 3-bp substitutions in the miR-640 target sites were constructed using the InFusion HD Cloning kit (Clonetech). HUVECs were co-transfected with one of these plasmids along with the control mimic, miR-640 mimic, control inhibitor, or miR-640 inhibitor. After a 48 h recovery period, cells were either lysed or cultured for 24 h in basal medium containing aliquots of Ang-1 (300 ng/mL) or PBS. Firefly and Renilla luciferase activities were measured with the Dual-Luciferase^®^ Reporter Assay System (Promega, Madison, WI, USA) and Firefly luciferase activity was normalized for Renilla activity to account for differences in transfection efficiencies.

### 2.14. Biotin-Labeled Pull-Down Assays

Biotinylated miR-640 pull-down assays with target mRNAs were performed as described earlier [23,24]. Briefly, HUVECs were transfected with biotinylated-control or biotinylated-miR-640 mimics (50 nM, Exqion Inc.). Forty eight hours later, cells were lysed with a hypotonic lysis buffer (100 mM KCl, 5 mM MgCl_2_, 20 mM Tris-Cl pH 7.5, 5 mM DTT, 0.3% NP-40, 60 U/mL RNase OUT, and 1× Complete Mini protease inhibitor (Roche, Basel, Switzerland)). Simultaneously, magnetic streptavidin beads (Dynabeads M-280 Streptavidin, #11205D, Invitrogen) were coated with bovine serum albumin (1 μg/μL) and yeast tRNA (1 μg/μL, Invitrogen) and incubated with supernatants while rotating at 4 °C for 2 h. Cell debris was cleared by centrifugation (≥10,000 *g* at 4 °C for 15 min) and cleared lysates were then incubated with the pre-coated beads rotating overnight at 4 °C. Beads were then washed with hypotonic lysis buffer and RNA was then released by adding 750 µL of TRIzol (Invitrogen) and 250 µL of RNase-free water. RNA was then precipitated using a standard chloroform-isopropanol method and was then subjected to reverse transcription and qPCR for detection of specific mRNA transcripts. 

### 2.15. Data Analysis

Data are expressed as means ± SEM. Differences between experimental groups were determined using a two-way analysis of variance followed by a Student–Newman–Keuls post-hoc test. *p* values < 0.05 were considered statistically significant.

## 3. Results

### 3.1. Regulation of miR-640 Expression

The effects of Ang-1 on miR-640 (mature form) expression were assessed in HUVECs maintained for 24, 48, and 72 h in basal medium containing PBS or Ang-1 (300 ng/mL). Ang-1 significantly decreased miR-640 expression at the three time points relative to PBS (Figure 1A). miR-640 precursor (pri-miR-640) and miR-640 levels were measured 2, 4, 6, and 12 h post PBS or Ang-1 exposure to assess whether Ang-1 inhibits miR-640 transcription. Relative to control, pri-miR-640 transcript levels remained unchanged 2, and 4 h post Ang-1 but then declined thereafter (Figure 1B). In comparison, expression of mature miR-640 increased significantly 2 h post Ang-1, but then decreased significantly 6, and 12 h post Ang-1 (Figure 1B).

Exosomes are mediators of intercellular communication and incorporate and transfer mRNAs and miRNAs. miR-640 levels in exosomes isolated from HUVECs pre-incubated with PBS or Ang-1 for 24 h were measured to evaluate whether increased excretion of miR-640 in the exosomes may account for the decrease in its levels in cells exposed to Ang-1. Exposure to Ang-1 for 24 h significantly decreased exosomal miR-640 levels relative to PBS suggesting that Ang-1-induced decrease in miR-640 levels was not due to increased excretion of this miRNA in the exosomes (Figure 1C).

The essential role of Tie-2 receptors in mediating the inhibitory effect of Ang-1 on miR-640 expression was assessed by infecting HUVECs with Ad-GFP (control) or Ad-Ex Tek (express recombinant soluble Tie-2 receptors). With Ad-GFP, Ang-1 significantly decreased miR-640 levels relative to PBS (Figure 1D). With Ad-Ex Tek, Ang-1 had no effect on miR-640 expression (Figure 1D). These observations indicate that activation of Tie-2 receptors is required for the inhibitory effect of Ang-1 on miR-640 expression.

The miR-640 gene is located within an intron of the GATA zinc finger containing 2A (GATAD2A) gene. GATAD2A mRNA and miR-640 levels were measured 2, 4, 6, and 12 h post PBS or Ang-1 exposure to assess whether Ang-1-induced decreases in miR-640 expression correlate with that of GATAD2A. Ang-1 significantly decreased GATAD2A mRNA levels after 6, and 12 h, time points at which miR-640 expression was also significantly decreased (Figure 1E).

HUVECs were pre-incubated for 24 h with PBS, Ang-2, FGF2, VEGF, or TGFβ to assess whether the inhibitory effect of Ang-1 on miR-640 expression is unique to this angiogenesis factor. Ang-2, VEGF, and TGFβ but not FGF-2 significantly decreased miR-640 relative to PBS (Figure 1F). It should be emphasized not all miRNAs whose expressions are downregulated by Ang-1 are also downregulated by other angiogenesis factors. For instance, Ang-1, VEGF, and FGF-2 decreased, Ang-2 increased, and TGF-β had no effect on miR-211-3p expression (Appendix A).

### 3.2. Effects of the miR-640 Mimic on Cell Survival, Migration, and Differentiation

HUVECs were transfected with control or miR-640 mimics. After a 48 h period of recovery, cells were maintained in complete medium, basal medium containing aliquots of PBS (control), or Ang-1 and survival, proliferation, migration, and differentiation were then measured. Transfection with the miR-640 mimic significantly increased miR-640 levels (Appendix A). With the control mimic, incubation with basal medium containing PBS decreased cell counts and increased cleaved caspase-3 levels relative to complete medium (Figure 2A,B). Cells incubated with basal medium containing Ang-1 had higher counts and lower cleaved caspase-3 levels relative to PBS (Figure 2A,B). These results confirmed that Ang-1 promotes EC survival and inhibits caspase-3 activation. With the miR-640 mimic, the effects of PBS and Ang-1 on cell counts and cleaved caspase-3 levels were similar to those measured with the control mimic (Figure 2A,B).

With the control mimic, Ang-1 increased EC migration and capillary-like tube formation relative to PBS (basal migration and differentiation; Figure 2C,D). With the miR-640 mimic, basal cell migration and capillary-like tube formation measured with PBS were significantly lower relative to those measured with the control mimic (Figure 2C,D). With the miR-640 mimic, Ang-1 failed to increase cell migration and capillary-like tube formation relative to PBS (Figure 2C,D).

These results suggest that overexpression of miR-640 inhibits the pro-differentiation and pro-migration effects of Ang-1 but has no effects on the pro-survival effects of Ang-1.

### 3.3. Effects of miR-640 Inhibitor on Cell Survival, Migration, and Differentiation

HUVECs were transfected with control (LNA-scrambled) or miR-640 (LNA-anti-miR-640) inhibitors to assess the functional importance of endogenous miR-640 on angiogenic processes. With control inhibitor, incubation with basal medium containing PBS significantly decreased cell counts and increased cleaved caspase-3 levels relative to the complete medium (Figure 3A,B). Incubation with basal medium containing Ang-1 increased cell counts and decreased cleaved caspase-3 levels relative to PBS (Figure 3A,B). With miR-640 inhibitor, the effects of basal medium containing PBS or Ang-1 on cell counts and cleaved caspase-3 levels were similar to those observed with the control inhibitor (Figure 3A,B). 

With the control inhibitor, Ang-1 significantly increased cell migration and capillary-like tube formation relative to PBS (Figure 3C,D). With the miR-640 inhibitor, basal cell migration and capillary-like tube formation measured with PBS increased significantly relative to values measured with the control inhibitor (Figure 3C,D). With the miR-640 inhibitor, Ang-1 significantly increased cell migration and capillary-like tube formation to levels that were higher than those measured with the control inhibitor (Figure 3C,D). 

These results suggest that endogenous miR-640 inhibits basal EC migration and differentiation and removal of this effect using the miR-640 inhibitor enhances the pro-migration and pro-differentiation effects of Ang-1.

### 3.4. miR-640 Directly Targets ZFP91

We used TargetScan (www.targetscan.org), miRSearch3.0 (www.exiqon.com/mirsearch), and miRanda (www.microrna.org) algorithms to identify potential targets of miR-640. Fifteen genes were commonly identified by the three algorithms with ZFP91 having the highest score (Figure 4A). ZFP91 had a relatively large 3′UTR (3854 bp), which contained three miR-640 binding regions located at nucleotides 125–131, 349–355, and 3096–3103 (Appendix A). We measured ZFP91 mRNA and protein levels in HUVECs transfected with miR-640 mimics and inhibitors to assess whether ZFP91 is a true target of miR-640. Transfection with the miR-640 mimic significantly decreased ZFP91 mRNA and protein levels (Figure 4B–D) while miR-640 inhibition significantly increased ZFP91 mRNA and protein levels (Figure 4B–F).

The binding of miR-640 to a luciferase reporter expressing a 2000 bp portion of ZFP91 3′UTR (wt) was assessed to confirm that miR-640 selectively targets ZFP91 mRNA. This portion is predicted to have two miR-640 binding regions (regions 1 and 2, Appendix A). Two reporters were constructed in which regions 1 (ZFP91-region1-mut) and 2 (ZFP91-region2-mut) were mutated (Figure 5). HUVECs were co-transfected with control or miR-640 mimics along with empty, ZFP91-wt, ZFP91-region1-mut, or ZFP91-region2-mut reporters (Figure 5A). Cells were allowed to recover for 48 h, lysed, and relative luciferase activities and protein levels were then measured. With the control mimic, ZFP91-wt reporter activity was significantly lower while those with ZFP91-region1-mut and ZFP91-region2-mut reporters were significantly higher than that of empty reporter (Figure 5B). With the miR-640 mimic, ZFP91-wt reporter activity decreased significantly relative to the control mimic (Figure 5B). This inhibitory effect of the miR-640 mimic was not evident when ZFP91-region1-mut and ZFP91-region2-mut reporters were transfected (Figure 5B). These results suggest that miR-640 binds to regions 1 and 2 of ZFP91 3′UTR. 

Pull-down assays in which biotinylated control or miR-640 mimics were transfected in HUVECs and were then pulled down were used to document direct binding of miR-640 to ZFP91 mRNA. The efficiency of the pull down assays was confirmed by detecting enriched miR-640 in the pull-down materials of cells transfected with the biotinylated miR-640 mimic (Figure 5C,D). ZFP91 mRNA levels were significantly enriched in the pull-down materials and were depleted in the lysate material of the cells transfected with the miR-640 mimic relative to the control mimic (Figure 5E,F). These results indicate that miR-640 directly binds to ZFP91 mRNA.

### 3.5. Regulation of ZFP91 Expression

HUVECs were pre-incubated for 12, and 24 h with PBS or Ang-1 and ZFP91 mRNA and protein levels were then measured to assess the effects of Ang-1 on ZFP91 expression. Ang-1 significantly increased ZFP91 mRNA and protein levels relative to PBS (Figure 6A–C). HUVECs were transfected with wild-type (ZFP91-wt) or mutated (ZFP91-region1-mut and ZFP91-region2-mut) 3′ UTR luciferase reporters to assess whether Ang-1 decreases the binding of endogenous miR-640 to ZFP91 3′ UTR. Cells were allowed to recover for 48 h after transfection with luciferase reporters, exposed to PBS or Ang-1 for 24 h, lysed, and relative luciferase reporter activities were measured. Ang-1 significantly increased ZFP91-wt luciferase reporter activity relative to PBS (Figure 6D). Ang-1 had no effects on ZFP91-region1-mut and ZFP91-region2-mut reporter activities (Figure 6D). These results suggest that Ang-1 decreased miR-640 binding to regions 1 and 2 of ZFP91 3′UTR presumably due to downregulation of miR-640 expression. 

HUVECs were exposed to PBS, Ang-2, FGF2, and VEGF for 24 h to investigate whether other angiogenesis factors other than Ang-1 regulate ZFP91 expression. Ang-2 and VEGF significantly increased ZFP91 protein levels relative to PBS while FGF2 had no effect (Figure 6E,F).

### 3.6. Role of ZFP91 in Angiogenesis

HUVECs were transfected with scrambled or ZFP91 siRNA oligos, allowed to recover for 48 h, incubated in the complete medium, basal medium containing PBS, or Ang-1, and angiogenic processes were then measured. Transfection with ZFP91 siRNA oligos significantly decreased ZFP91 expression (Appendix A). With scrambled siRNA oligos, incubation with the basal medium containing PBS significantly decreased cell numbers and increased caspase-3 levels relative to the complete medium (Figure 7A,B). Incubation with the basal medium containing Ang-1 significantly increased cell numbers and decreased cleaved caspase-3 levels relative to PBS (Figure 7A,B). With ZFP91 siRNA, the effects of Ang-1 on cell numbers and cleaved caspase-3 levels were similar to those observed with scrambled siRNA (Figure 7A,B) suggesting that ZFP91 does not play a major role in the pro-survival effects of Ang-1.

With scrambled siRNA, Ang-1 increased cell migration and capillary-like tube formation relative to PBS (Figure 7C,D). With ZFP91 siRNA, basal cell migration and capillary-like tube formation measured with PBS were significantly lower than those measured with scrambled siRNA (Figure 7C,D). With ZFP91 siRNA, Ang-1 failed to increase cell migration and capillary-like tube formation relative to PBS (Figure 7C,D). These results indicate that ZFP91 is required for the pro-migration and pro-differentiation effects of Ang-1.

## 4. Discussion

The main findings of this study are: (1) Ang-1 induced Tie-2-dependent decreases in miR-640 levels; (2) miR-640 inhibited pro-migration and pro-differentiation effects of Ang-1 but had no effects on cell survival; (3) ZFP91 was a direct target of miR-640; (4) Ang-1 upregulated ZFP91 expression by decreasing miR-640 binding to ZFP91 mRNA; and (5) ZFP91 was required for Ang-1-induced cell migration and capillary-like tube formation.

### 4.1. Regulation of miR-640 Expression

The expression and functional roles of miR-640 in various cellular processes are poorly documented. miR-640 expression is elevated in cholangiocarcinoma, ovarian carcinoma, and lymphocytic leukemia cells [25,26,27]. Dong et al. [28] reported that miR-640 expression is elevated in degenerative intervertebral tissues of humans and that pro-inflammatory cytokines such as the tumor necrosis factor (TNF-α) and interleukin-1 (IL-1β) induce miR-640 expression in nucleus pulposus cells. In the present study, we reported for the first time that miR-640 levels in ECs decreased significantly after several hours of exposure to Ang-1 and that this response was mediated through a Tie-2-dependent decrease in transcription of miR-640 precursor. We also found that miR-640 expression decreased significantly 24 h after Ang-2 exposure (Figure 1F). Ang-2 binds Tie-2 receptors with equal affinity to that of Ang-1, although Ang-2 elicits only weak Tie-2 receptor phosphorylation and can competitively inhibit Ang-1-induced Tie-2 phosphorylation [2]. There is also evidence that even at relatively low concentrations; Ang-2 is capable of activating the pro-survival and pro-migration PI3 kinase/AKT pathway and of promoting EC survival, migration and differentiation [29]. In the current study, the decline in miR-640 in response to 300 ng/mL of Ang-2 suggests that activation of Tie-2 and its downstream signaling pathways by Ang-2 was sufficiently strong to trigger significant downregulation of miR-640 expression in a fashion similar to that triggered by Ang-1.

miR-640 is located within an intron of the GATAD2A protein coding gene. We assessed whether Ang-1 regulates miR-640 and GATAD2A expression in a similar fashion since many intronic miRNAs are regulated in a similar fashion to their host genes. Ang-1 decreased GATAD2A mRNA levels in a similar time frame (6 and 12 h) to that of miR-640 (Figure 1). These results suggest that common signaling pathways may have been utilized by the Ang-1/Tie-2 axis to downregulate GATAD2A and miR-640 expression. GATAD2A is a subunit of the NuRD (nucleosome remodeling and histone deacetylation) complex, which plays key roles in general repression of transcription, cellular differentiation, and embryonic development [30]. Regulation of GATAD2A transcription has not been extensively studied. We speculated that the Ang-1/Tie-2 pathway downregulates miR-640 and GATAD2A expression through three mechanisms. First, Dong et al. [28] identified several NFκB binding sites in the human GATAD2A promoter and showed that pro-inflammatory cytokines such as TNF-α upregulate miR-640 and GATAD2A expression in nucleus pulposus cells through NFκB. Our group and others have previously reported that the Ang-1/Tie-2 pathway inhibits NFκB activation in ECs through two distinct mechanisms [31,32]. It is possible that downregulation of miR-640 and GATAD2A expression by Ang-1 might have been mediated through inhibition of basal NFκB activity. Second, Zhou et al. [17] reported that hydrogen sulfide decreases miR-640 expression in ECs through the mammalian target of rapamycin (mTOR) and VEGF2 receptor pathways. The fact that the Ang-1/Tie-2 pathway strongly activates mTOR in ECs [22] suggests that this pathway may be involved in the downregulation of miR-640 and GATAD2A expression by Ang-1. Third, Ang-1/Tie-2 activates the early growth response 1 (EGR1) transcription factor in ECs [33]. Analysis of the GATAD2A promoter revealed enriched binding sites for EGR1 suggesting that this transcription factor might be involved in the inhibitory effect of the Ang-1/Tie-2 pathway on miR-640 and GATAD2A expression. We should emphasize that the levels of mature miR-640 increased significantly while those of pri-miR-40 remained unchanged in response to 2 h exposure to Ang-1 (Figure 1B) suggesting that acute exposure to Ang-1 does not alter the transcription of miR-640. Future studies are clearly warranted to identify exact mechanisms involved in the regulation of miR-640 expression by the Ang-1/Tie-2 pathway.

### 4.2. Regulation of Angiogenesis by miR-640

Little information is available regarding the physiological roles of miR-640. In ovarian cancer, high levels of miR-640 are associated with better survival rates [26]. In nucleus pulposus (NP) cells, miR-640 promotes pro-inflammatory cytokine expression by selectively targeting low-density lipoprotein receptor-related protein-1 (LRP1), an inhibitor of NFκB transcription factor [28]. It has also been shown that miR-640 promotes senescence of NP cells by inhibiting the WNT signaling pathway [28]. To our knowledge, the only study linking miR-640 to angiogenesis is that of Zhou et al. [17] who reported that miR-640 inhibits hydrogen sulfide-induced EC migration and differentiation by selectively targeting of hypoxia-induced factor 1α (HIF1α).

The current study demonstrated that miR-640 has anti-angiogenic properties, as indicated by the inhibitory effects of its mimic and the enhancing effects of its inhibitor on EC migration and differentiation. The mechanisms behind miR-640 anti-angiogenic effects remain unclear. In the current study, we identified miR-640 intracellular targets using three algorithms commonly used for predicting miRNA targets. ZFP91 is considered one of the top targets of miR-640 because of the presence of three putative miR-640 binding regions in the ZFP91 3′UTR. Here we provide three sets of observations that strongly support the notion that ZFP91 is a direct target of miR-640 in ECs. First, the miR-640 inhibitor increased while the miR-640 mimic decreased ZFP91 mRNA and protein levels. Second, luciferase reporter activity of 2000 bp of wild type ZFP 3’UTR decreased significantly by the miR-640 mimic. This inhibitory effect was not detected when miR-640 binding regions 1 (nucleotide 112–121) and 2 (nucleotide (335–347) of the ZFP91 3’UTR were mutated indicating that miR-640 binds to these two regions and destabilizes ZFP91 mRNA. Third, ZFP91 mRNA was significantly enriched in the pull-down materials of the biotinylated miR-640 mimic indicating that miR-640 directly binds to ZFP91 mRNA.

### 4.3. ZFP91 and Ang-1-Induced Angiogenesis

ZFP91 is an atypical ubiquitin E3 ligase that reportedly activates NF-κB-inducing kinase (NIK) via Lys^63^-linked ubiquitination in the non-canonical NF-κB signaling pathway [34]. ZFP91 has been described as a promoter of carcinogenesis in prostate cancer and myelogenous leukemia [35,36]. It is unclear whether biological functions of ZFP91 are solely achieved by its E3 ligase activity and activation of NIK or through yet to be determined mechanisms. The fact that ZFP91 has a secondary structure typical of transcription factors, i.e., five zinc-finger domains, one leucine-zipper pattern, one coiled-coil structure, and several nuclear localization signals, suggests that it may function as a transcription factor [37].

The role of ZFP91 in angiogenesis is unknown. In the present study, we reported that ZFP91 expression in ECs was significantly induced by Ang-1, Ang-2, and VEGF. In the case of Ang-1, upregulation of ZFP91 appeared to be mediated through downregulation of miR-640 expression and attenuation of the binding of this miRNA to two regions of the ZFP91 3’UTR (Figure 6). Our study also indicates that Ang-1-induced EC migration and differentiation required the presence of ZFP91. These results indicate that Ang-1 promotes angiogenesis, more specifically cell migration and capillary-like tube formation, by repressing miR-640 expression and alleviation of its inhibitory effect on ZFP91 expression. This is the first evidence of a pro-angiogenic role for ZFP91. The mechanisms through which ZFP91 promotes EC migration and differentiation remain unclear. One possible mechanism is activation of NIK and the non-canonical NFκB pathway [34]. This pathway promotes differentiation of ECs derived from rheumatoid arthritis tissues and cancer tissues [38]. It has been proposed that the non-canonical NFκB pathway promotes cell migration through increased production of interleukin-8 (IL-8), chemokine receptor CXCR4, and metal metalloproteinase 9 [39]. Activation of the non-canonical NFκB pathway in ECs also increases production of the pro-angiogenic CXC12 chemokine (CXCR4 ligand) [40]. It should be emphasized IL-8 strongly stimulates EC migration and differentiation downstream from Ang-1/Tie-2 axis and that CXCR4 expression in ECs is upregulated by Ang-1 [20,22]. ZFP91 may also regulate EC migration and differentiation through direct interactions with ARF (cyclin-dependent kinase inhibitor 2A, isoform 4) and which exerts negative effects on the cell cycle through p53-dependent and independent mechanisms [41]. Another pathway through which ZFP91 may promote angiogenesis is hypoxia-induced factor 1α (HIF1α), which is strongly upregulated by ZFP91 in colon cancer cells [42]. More recently, ZFP91 reportedly activates mitogen-activated protein kinases (MAPKs), particularly the ERK1/2 pathway in macrophages [30]. The ERK1/2 pathway promotes the pro-migration and differentiation effects of Ang-1 in ECs [6]. Finally, it has been recently revealed that ZFP91 directly ubiquitinates heterogeneous nuclear ribonucleoprotein F (hnRNP F) at Ly 185 and degrades it through the proteasome pathway [43]. HnRNP F is a member of hnRNP family of RNA-binding protein that functions in the regulation of nucleic acid metabolism such as transcription, translation, and alternative splicing. The functional involvement of the non-canonical NFκB, ARF, HIF1α, ERK1/2, and hnRNP F in the pro-angiogenic effects of ZFP91 in ECs remains to be investigated.

### 4.4. Other Targets of miR-640

Figure 4 shows that TargeScan, miRsearch, and miRanda algorithms predict several genes other than ZFP91 as targets of miR-640. Several of the predicted genes such as CCNG1 (Cyclin G1), CCNA1 (Cyclin A1), CD44, and NELL1 play significant roles in the regulation of cellular survival, migration, and differentiation. For instance, coordinated activation of the cell cycle regulators CCNG1 and CCNA1 with cyclin-dependent protein kinases is required for normal regulation of cell proliferation and locomotion [44]. The multifunctional hyaluronan receptor CD44 is expressed at relatively high levels on the surface of ECs and knocking down its expression results in impairment of EC differentiation and upregulation of several pro-inflammatory chemokines [45]. NELL1, also known as Neuropilin 1 (NRP1), is a cell surface receptor that binds to VEGF and SEMA3A. NELL1 plays a critical role in embryonic angiogenesis and enhances angiogenesis in adult vasculature [46]. Whether CCNG1, CCNA1, CD44, and NELL1 are directly targeted by miR-640 and whether direct targeting of these genes mediates the anti-angiogenic properties of miR-640 need to be investigated in future studies.

In summary, we reported that miR-640 directly targeted ZFP91 and that exposure of ECs to Ang-1 elicited upregulation of ZFP91 expression through transcriptional down-regulation of miR-640. We also identified important roles for ZFP91 protein in the regulation of Ang-1-induced endothelial cell migration and differentiation. 

## Figures and Tables

**Figure 1 cells-09-01602-f001:**
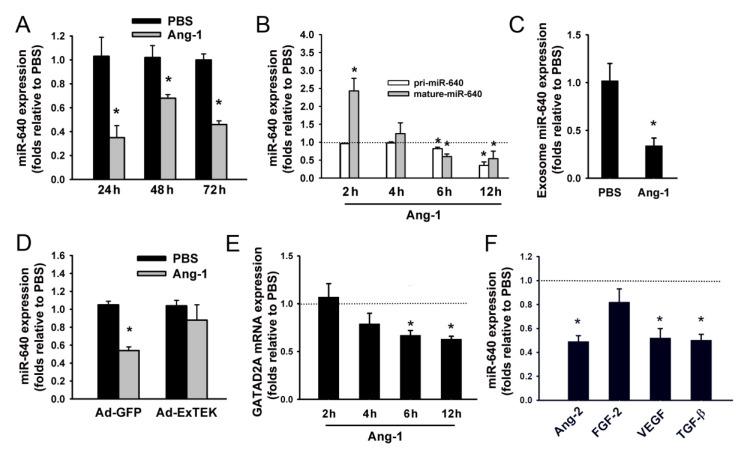
Regulation of miR-640 expression. (**A**): Expression of mature miR-640 in human umbilical vein endothelial cells (HUVECs) exposed to PBS (control), or Ang-1 for 24, 48, and 72 h. Values are means ± SEM, expressed as folds relative to PBS. * *p* < 0.05, compared to PBS. (**B**): Expression of pri-miR-640 and mature miR-640 in HUVECs exposed to PBS, or Ang-1 for 2, 4, 6, and 12 h. * *p* < 0.05, compared to PBS. (**C**): miR-640 levels in HUVEC exosomes in response to 24 h exposure to PBS, or Ang-1. * *p* < 0.05, compared to PBS. (**D**): miR-640 expression in HUVECs infected with Ad-GFP or Ad-ExTEK and then exposed to PBS or Ang-1 for 24 h. * *p* < 0.05, compared to PBS. (**E**): Expression of GATAD2A in HUVECs exposed to PBS, or Ang-1 for 2, 4, 6, and 12 h. Values are expressed as expressed as folds relative to PBS. * *p* <0.05, compared to PBS. (**F**): Expression of miR-640 in HUVECs exposed to PBS, Ang-2, FGF-2, VEGF, or TGF-β. Values are expressed as folds relative to PBS. * *p* < 0.05, compared to PBS.

**Figure 2 cells-09-01602-f002:**
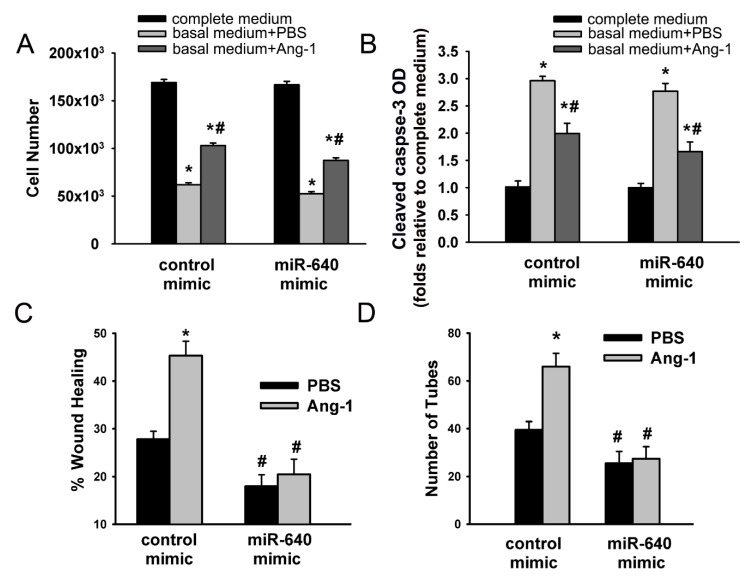
Effects of the miR-640 mimic on cell survival, migration, and differentiation. (**A** and **B**): HUVECs were transfected with control or miRNA 640 mimics. Equal numbers of cells were then maintained in complete (20% FBS), basal medium (2% FBS) containing PBS, or Ang-1 (300 ng/mL). Cell numbers and cleaved caspase-3 optical densities (OD) were evaluated 24 h later. Values are means ± SEM. * *p* < 0.05, compared to cells maintained in complete medium. # *p* < 0.05, compared to PBS. (**C**): Scratch wound healing in HUVECs transfected with control or miR-640 mimics and cultured in basal medium containing PBS or Ang-1. Percentage wound healing was measured 8 h after wounding with a pipette tip. Values are means ± SEM. * *p* < 0.05, compared to PBS. # *p* < 0.05, compared to cells transfected with the control mimic and treated with PBS or Ang-1. (**D**): Total tube numbers of HUVECs transfected with control or miR-64 mimics and maintained for 24 h in plates precoated with growth factor-reduced Matrigel and basal medium containing PBS or Ang-1. Values are means ± SEM. * *p* < 0.05, compared to PBS. # *p* < 0.05, compared to corresponding cells transfected with the control mimic and treated with PBS or Ang-1.

**Figure 3 cells-09-01602-f003:**
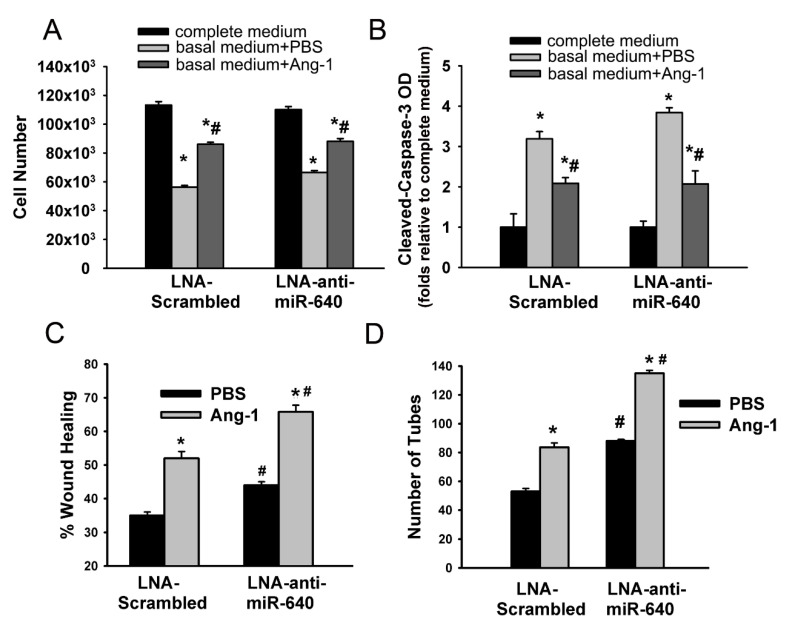
Effects of the miR-640 inhibitor on cell survival, migration, and differentiation. (**A** and **B**): HUVECs were transfected with control (LNA-scrambled) or miR-640 (LNA-anti-miR-640) inhibitors. Equal numbers of cells were cultured in complete, and basal medium containing PBS or Ang-1. Cell counts and cleaved caspase-3 optical densities (OD) were evaluated 24 h later. Values are means ± SEM. * *p* < 0.05, compared to cells maintained in complete medium. # *p* < 0.05, compared to PBS. (**C**): Scratch wound healing assays of HUVECs transfected with control or miR-640 inhibitors. Cells were maintained in basal medium containing PBS or Ang-1. Percentage wound healing was measured 8 h after wounding with a pipette tip. Values are means ± SEM. * *p* < 0.05, compared to PBS. # *p* < 0.05, compared to cells transfected with control inhibitor and treated with PBS or Ang-1. (**D**): Total tube numbers of HUVECs transfected with control or miR-64 inhibitors and maintained for 24 h in plates pre-coated with growth factor-reduced Matrigel and basal medium containing PBS or Ang-1. Values are means ± SEM. * *p* < 0.05, compared to PBS. # *p* < 0.05, compared to cells transfected with control inhibitor and treated with PBS or Ang-1.

**Figure 4 cells-09-01602-f004:**
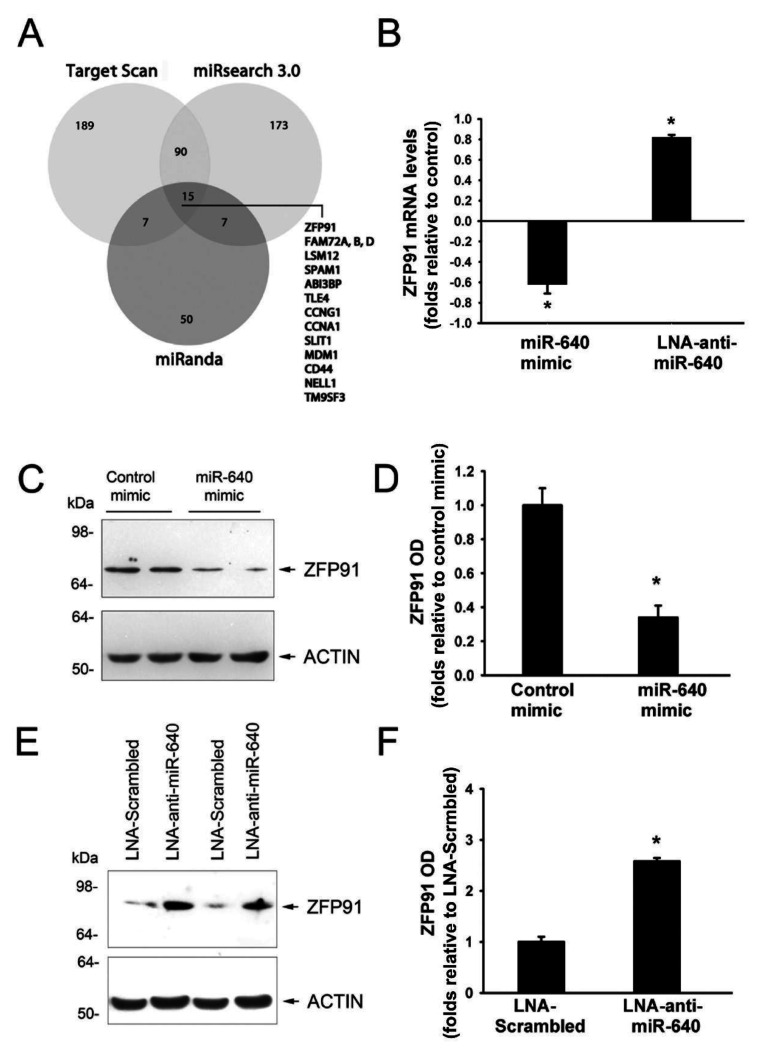
Identification of miR-640 targets. (**A**): Venn diagram displaying computationally predicted targets of hsa-miR-640 by TargetScan, miRanda, and miRsearch3.0. Commonly predicted targets by the three algorithms are listed. (**B**): ZFP91 mRNA expression in HUVECs transfected 48 h earlier with the control mimic, miR-640 mimic, control inhibitor, or miR-640 inhibitor. Values are means ± SEM. * *p* < 0.05, compared to the corresponding control group (mimic or inhibitor). (**C** and **D**): Representative immunoblots and optical densities (OD) of ZFP91 protein in HUVECs transfected 48 h earlier with control or miR-640 mimics. Values are means ± SEM, expressed as folds relative to the control mimic. * *p* < 0.05, compared to the control mimic. (**E** and **F**): Representative immunoblots and optical densities (OD) of the ZFP91 protein in HUVECs transfected 48 h earlier with control or miR-640 inhibitors. Values are means ± SEM and are expressed as folds relative to control inhibitor. * *p* < 0.05, compared to control inhibitor.

**Figure 5 cells-09-01602-f005:**
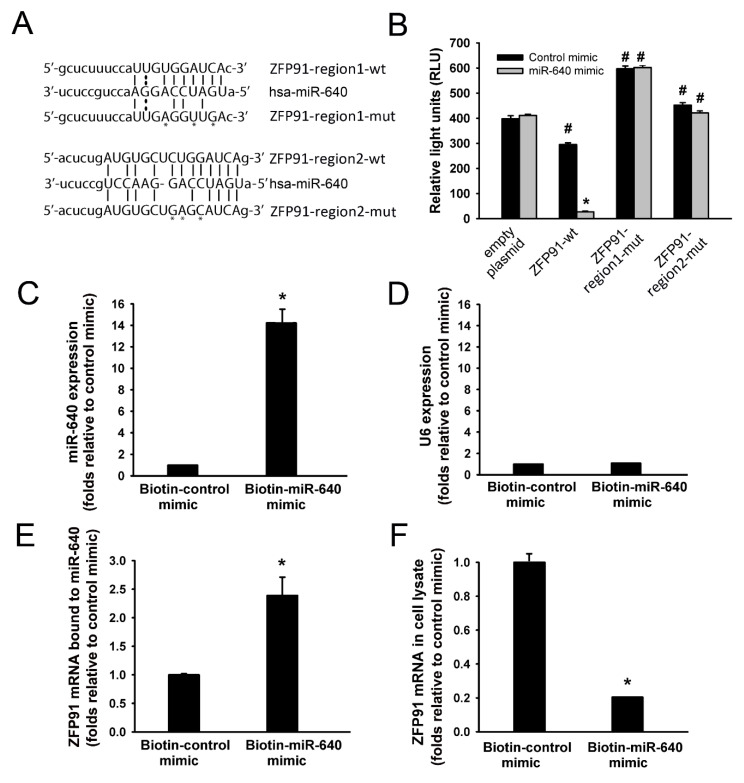
ZFP91 is a direct target of miR-640. (**A**): Sequence alignment of miR-640 and its target sites in 3′UTRs of ZFP91. (**B**): Relative luciferase activity in HUVECs transfected with control or miR-640 mimics and luciferase reporter plasmid expressing wild type (wt) or mutated 3′UTRs of ZFP91. The mutated 3′UTRs of ZFP91 have mutated miR-649 binding regions 1 or 2. Values are means ± SEM, expressed as relative light units (RLU). * *p* < 0.05, compared to the control mimic. # *p* < 0.05, compared to empty plasmid. (**C** and **D**): Levels of miR-640 and U6 RNA measured 48 h after transfection of HUVECs with biotinylated control or mi-R-640 mimics. * *p* < 0.05, compared to the control mimic. (**E** and **F**): Levels of ZFP91 mRNA in the pulled down materials of biotinylated control and miR-640 mimics. Values are means ± SEM, expressed as folds relative to the control mimic. * *p* < 0.05, compared to the control mimic. (**F**): Levels of ZFP91 mRNA in the supernatant of cell lysates following pull downs of biotinylated control or miR-640 mimics. Values are means ± SEM, expressed as folds relative to the control mimic. * *p* < 0.05, compared to the control mimic.

**Figure 6 cells-09-01602-f006:**
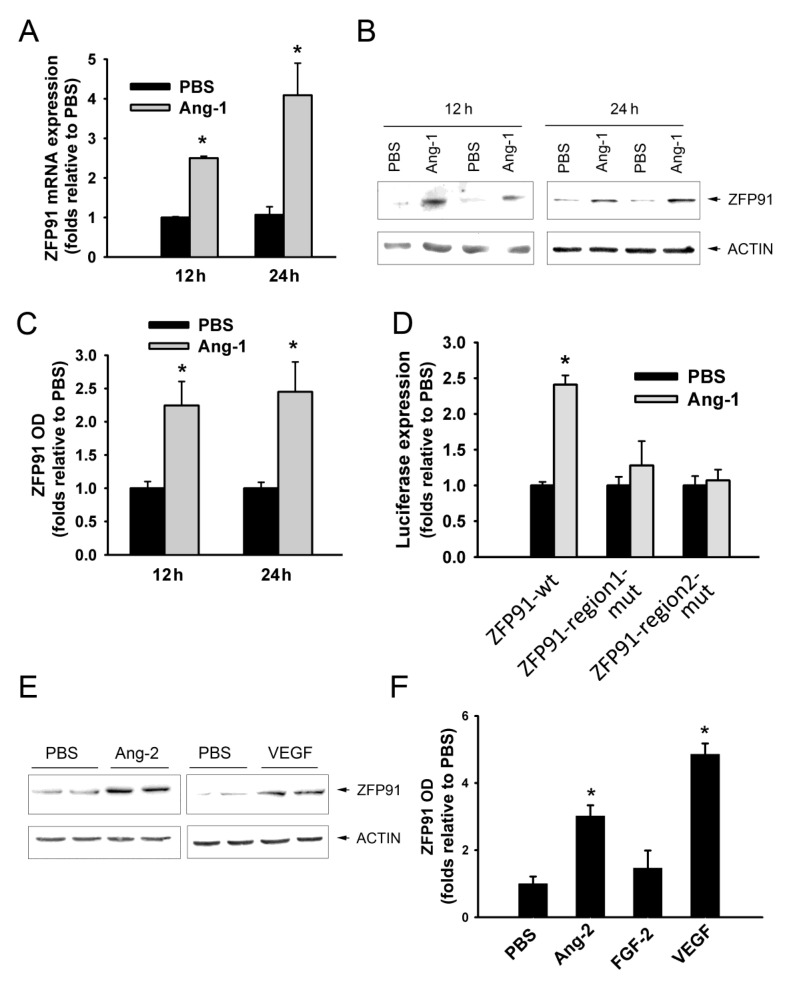
Regulation of ZFP91 expression by Ang-1. (**A**): Expression of ZFP91 mRNA in HUVECs exposed to PBS (control), or Ang-1 for 12, and 24 h. Values are means ± SEM, expressed as folds relative to PBS. * *p* < 0.05, compared to PBS. (**B** and **C**): Representative immunoblots and optical densities (OD) of ZFP91 protein in HUVECs exposed to PBS, or Ang-1 for 12, and 24 h. * *p* < 0.05, compared to PBS. (**D**): Relative luciferase activity in HUVECs transfected with luciferase reporter plasmid expressing wt or mutated 3’UTRs of ZFP91 then exposed to PBS, or Ang-1 for 24 h. * *p* < 0.05, compared to PBS. (**E** and **F**): Representative immunoblots and optical densities (OD) of ZFP91 protein in HUVECs exposed to PBS, Ang-2, FGF-2, or VEGF for 24 h. Values are expressed as folds relative to PBS. * *p* < 0.05, compared to PBS.

**Figure 7 cells-09-01602-f007:**
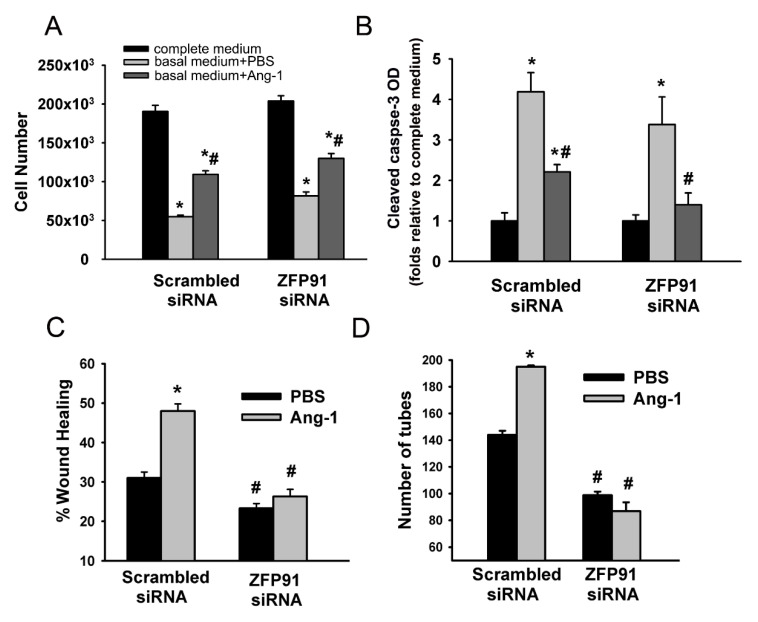
Effect of ZFP91 inhibition on Ang-1-induced angiogenesis. (**A** and **B**): HUVECs were transfected with scrambled or ZFP91 siRNA oligos. Equal numbers of cells were then maintained in the complete medium (20% FBS), basal medium (2% FBS) + PBS, or basal medium + Ang-1 (300 ng/mL). Cell counts and cleaved caspase-3 optical densities (OD) were measured 24 h later. * *p* < 0.05, compared to complete medium. # *p* <0.05, compared to PBS. (**C**): Percentage of wound healing in HUVECs transfected with scrambled or ZFP91 siRNA oligos. * *p* < 0.05, compared to PBS. # *p* < 0.05, compared to cells transfected with scrambled siRNA oligos and treated with PBS or Ang-1. (**D**): Total tube numbers of HUVECs transfected with scrambled or ZFP91 siRNA oligos and maintained for 24 h in plates pre-coated with growth factor-reduced Matrigel and basal medium containing Ang-1 or PBS. Values are means ± SEM. * *p* < 0.05, compared to PBS. # *p* < 0.05, compared to cells transfected with scrambled siRNA oligos and treated with PBS or Ang-1.

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
