# Peer review of "Roles of miR-640 and Zinc Finger Protein 91 (ZFP91) in Angiopoietin-1-Induced In Vitro Angiogenesis"

_cells, 2020, doi:10.3390/cells9071602_

Round 1

Reviewer 1 Report

In the current study the authors are addressing the mechanisms regulating Ang1-mediated endothelial cell (EC) survival as well as cell migration and pro-angiogenic – tubule capillary-like formation.  Herein, the authors report that miR-640 is an endogenous repressor of EC pro-angiogenic status. Under Ang1 challenge, miR-640 is downregulated, leading to ZFP91 mRNA overexpression and Ang1-mediated pro-angiogenic activities on cultured ECs.

The study is well-designed, and the conclusions associated to Ang1-mediated biological activities are supported by the results. In itself, it is sufficient for publication – at least, this is the opinion of the current reviewer.

However, the authors are providing challenging data, and the authors are strongly encouraged to develop (to propose) in depth potential explanations.

For instance:

  • Why an increase of miR-640 levels (thus, expected down-regulation of ZFP19 mRNA expression) is not affecting cell viability and caspase-3 activity as opposed to wound healing and tubule capillary-like formation. The candidates should elaborate.
  • Ang-2, VEGF, TGF-beta and FGF-2 are equipotent to Ang1 to downregulating miR-640 expression. Do they have an example where other growth factors do not down regulate miR-640?
  • Although numerous studies are reporting that Ang2/Tie2 complex can have agonistic activities; one might argue that under numerous conditions, Ang2 is considered as an endogenous Tie2 antagonist. The authors should elaborate on this observation.
  • Based on Fig 1F; and looking at Fig 6A and F, we observe that Ang2 and VEGF are about equipotent to Ang1 to promoting ZFP91 mRNA expression, whereas FGF-2 is inefficient. Considering that FGF-2 did reduce miR-640 expression, one might expect to see similar capacity of FGF-2 to overexpress ZPF91 mRNA. The authors should discuss about this observation. This is even more important considering that FGF-2 is a potent inducer of ECs viability, migration and angiogenesis.

Reviewer 2 Report

This very interesting study reports a new mechanism of Ang-1 mediated angiogenesis.  Ang-1 down regulation of miR-640 in HUVECs, which targets ZFP91, leads to increased expression of ZFP91.  Although the mechanisms remain undefined, the authors clearly defined that ZFP91 promotes endothelial cell migration and tube formation. Overall, the conclusion of this manuscript is supported by well-designed experiments and consistent results.

One weakness of this manuscript was the lack of experiments to address the specificity of Ang-1 mediated down regulation of miR-640 and the mechanisms involved. Several factors including Ang-1 antagonist Ang-2 also down regulate miR-640.  In the discussion, the authors suggested that common pathways such as NF-kB and mTOR may be involved. This can be easily addressed as specific inhibitors for each pathway are available.

Minor errors, problems, and suggestions:

1. Line 98. the term “Adenoviral Transfection” is inaccurate. Introducing gene expression by means of viral vectors is referred to as "transduction", not "transfection". This can be changed to either “Adenoviral Transduction” or “ Adenoviral infection”

2. Fig. 1. The levels of pri-miR-640 and miR-640 in Ang-1 treated HUVECs were compared relative to that of HUVECs treated with PBS. In both figure legend and Y-axis labeling, the authors used  “folds from PBS”, which is very confusing. This same wording is used throughout the manuscript.  Better wording such as "Folds vs. PBS" or "Folds relative to PBS" is preferred.    

3. Line 187-189: There is an inconsistency between the presented data in Fig. 1B and the statement in the text.  In the text, it says, "Relative to control, pri-miR-640 transcript levels significantly increased after 2 h then declined significantly 6, and 12 h post Ang-1 (Figure 1B). In comparison, miR-640 levels decreased significantly 6, and 12 h post Ang-1 exposure 189 (Figure 1B).   In the figure, it showed that the pri-miR-640 did not change much at 2h and 4h but slightly decreased at 12h. Rather, miR-640 significantly increased at 2h.  This needs to be corrected.  In addition, the time point for experiment described in Fig. 1C was not mentioned.  Given that the levels of miR-640 are higher at 2h post-Ang-1 stimulation, it is important to measure the levels of miR-640 in exosomes at both early time points (before 12h) and late time points (24h).  In Fig.1E, the expression pattern of GATAD2A is consistent with that of pri-miR-640 shown in Fig. 1B, further suggesting that Ang-1 does not affect transcription of pri-miR-640 during early time pints but somehow increases the level of mature miR-640 at 2h.

4. In Fig. 3A, it shows that miR-640 targets a number of factors with ZFP91 as a top target.  The authors did not mention any of the other targets with regards to their possible roles in angiogenesis.  Interestingly, it is well documented that CD44 is a multistructural and multifunctional cell surface molecule involved in cell proliferation, cell differentiation, cell migration, angiogenesis, presentation of cytokines, chemokines, and growth factors to the corresponding receptors, and docking of proteases at the cell membrane, as well as in signaling for cell survival.  This can be discussed in Discussion and merits further investigation.

5). Although FGF2 down regulates miR-640 (Fig. 1F), it has little effect on ZFP91 expression (Fig. 6F).  This inconsistency needs to be addressed or discussed.  Based on the graph, it seems that BGF2 does slightly increase ZFP91 expression. Conducting this experiment with additional repeats may be helpful.     
